# Healing Patterns of Non-Collagenated Bovine and Collagenated Porcine Xenografts Used for Sinus Floor Elevation: A Histological Study in Rabbits

**DOI:** 10.3390/jfb13040276

**Published:** 2022-12-05

**Authors:** Yuhei Miyauchi, Takayuki Izutani, Yuki Teranishi, Takahisa Iida, Yasushi Nakajima, Samuel Porfirio Xavier, Shunsuke Baba

**Affiliations:** 1Department of Oral Implantology, Osaka Dental University, 8-1 Kuzuhahanazonocho, Hirakata, Osaka 573-1121, Japan; 2ARDEC Academy, Viale Giovanni Pascoli 67, 47923 Rimini, Italy; 3Department of Oral and Maxillofacial Surgery and Periodontology, Faculty of Dentistry of Ribeirão Preto, University of São Paulo, Ribeirão Preto 14040-904, São Paulo, Brazil

**Keywords:** bone substitutes, sinus floor augmentation, bone formation, bone regeneration, bone resorption, animal model

## Abstract

Objective: To compare healing of collagenated and non-collagenated xenografts used for maxillary sinus floor elevation. Materials and Methods: Two different xenografts were used: deproteinized bovine bone (DBBM group) and collagenated corticocancellous porcine bone (collagenated group). Healing was studied after 2, 4, and 8 weeks. The loss of dimensions of the elevated area and the percentages of new bone, xenograft remnants, osteoclastic zones, vessels, inflammatory infiltrates, and soft tissues were analyzed. Three regions were evaluated: close to the bone walls (bone wall region), subjacent the sinus mucosa (submucosa region), and the center of the elevated area (middle region). The primary variables were the percentage of new bone and xenograft remnants. Results: Between 2 and 8 weeks, the elevated areas showed a reduction of 16.3% and 52.2% in the DBBM and collagenated groups, respectively (*p* < 0.01 between the two areas after 8 weeks). After 8 weeks, the highest content of new bone was observed in the bone wall region, which was higher in the collagenated group than in the DBBM group (41.6% and 28.6%, respectively; *p* < 0.01). A similar quantity of new bone was found between the two groups in other regions. A higher percentage of vessels in all regions evaluated (*p* < 0.01) and soft tissue in the sub-mucosa region (*p* < 0.05) was found in the collagenated group than in the DBBM group. Conclusions: The present study showed that both xenografts allowed new bone formation. In comparison with the non-collagenated xenograft, the collagenated xenograft underwent higher resorption, resulting in greater shrinkage of the elevated space after sinus lifting and a higher content of new bone in the regions close to the bone walls. Clinical relevance: In this study, the region adjacent to the bone wall showed the highest new bone content. This region resembles the base of the sinus, closest to the sinus floor and walls, and is the most important region from a clinical point of view because it is where the implant will be installed. Residues of the biomaterial remained after 8 weeks of healing. Other reports have shown that these biomaterial residues may interfere with the integration of implants.

## 1. Introduction

The presence of sufficient bone volume is a necessary prerequisite for implant installation and to provide a long-term successful outcome [1]. When no sufficient bone volume is available in the posterior region of the maxilla, sinus floor elevation has been shown to be a reliable surgical approach that allows oral rehabilitation using implants [2,3]. Lateral access is one of the most common approaches used for sinus lifting [2].

A series of anatomical characteristics should be considered before performing a surgical procedure. Among them, we should consider the height of the residual bone crest, the angle and position of the palatal-nasal recess in relation to the floor of the sinus, the angle of the floor sinus formed by the lateral and medial sinus walls, the width of the lateral wall, the position of the posterior superior alveolar artery, the presence of intra-sinus pathologies and septa, and the patency of the antrum [4,5].

The space gained after sinus floor elevation tends to be lost over time [6,7,8]. Volumetric changes after sinus lifting evaluated using computed tomography (CTs) scans or cone beam CTs (CBCTs) were reported in a systematic review [9]. To maintain the elevated space, biomaterials [10,11,12,13], implants without filler material [3,13,14,15,16,17,18,19,20,21], or resorbable or non-resorbable devices [22,23,24,25], were placed within the elevated space.

Histometric outcomes of various grafting materials used for sinus floor elevation have been discussed in various systematic reviews [10,11,12]. A systematic review with network meta-analyses [10] found differences in bone formation and graft resorption among the various types of xenografts included in the assessments. A systematic review with a meta-analysis [12] concluded that autogenous bone was the best choice when a high content of newly formed bone was needed. Furthermore, other biomaterials might be considered reliable bone substitutes in cases of concern regarding the donor site. The outcomes from the review showed significantly greater new bone formation with bovine bone than with hydroxyapatite alone, and better results with a biphasic graft composed of β-tricalcium phosphate and hydroxyapatite than with bovine bone. In a Bayesian network meta-analysis [11], however, it was concluded that autogenous bone showed reduced new bone formation than several other biomaterials used as grafts. 

Different healing features and resorption rates of different biomaterials have also been reported in rabbits experiments [26,27,28,29]. In one of these experiments [26], sinus augmentation was performed either with deproteinized bovine bone mineral (DBBM) or with a collagen sponge. Healing was evaluated after different periods and showed stable dimensions of the elevated space between 7 and 40 days, while at the collagen site, >50% of the dimensions were lost. The total amount of new bone was higher at DBBM sites than at collagen sponge sites. 

In a similar study in rabbits [27], sinus augmentation was performed using either DBBM or autogenous bone, and simultaneous implants were installed. Healing was evaluated after 7 and 40 days. After 40 days of healing, a higher percentage of new bone was found at autogenous sites than at DBBM sites. However, two thirds of the area of the elevated space was lost at the autogenous sites, while the dimensions were maintained at the DBBM sites.

In another similar study [28], sinus floor augmentation was performed bilaterally using collagenated corticocancellous porcine bones. At the test site, a collagen membrane was placed adjacent to the sinus mucosa prior to grafting. Similar amounts of new bone were found in both the sinuses. However, approximately 50% of the elevated space was lost during healing.

These studies showed similar percentages of new bone [26,28] but a higher loss in the dimension of the elevated space in the collagenated group than in the non-collagenated group. A comparison between these two different xenografts on different parameters, such as bone formation, graft resorption, vessel formation, presence of resorptive zones, and inflammatory infiltrates, might still provide useful information. Moreover, detailed data on these parameters in different regions of the elevated regions, such as those close to the bone walls or sinus mucosa, might allow the disclosure of differences in healing between the two biomaterials. Hence, the aim of the present study was to compare the healing of collagenated and non-collagenated xenografts used for maxillary sinus floor elevation.

The hypothesis of this study was that collagenated and non-collagenated xenografts used for sinus floor augmentation might result in different percentages of newly formed bone.

## 2. Materials and Methods

### 2.1. Ethical Statement

The protocols for both experiments were approved by the Ethical Committee of the Faculty of Dentistry in Ribeirão Preto of the University of São Paulo (USP, São Paulo, Brazil). The protocol numbers were 2017.1.278.58.9 [30] and 2015.1.834.58.7 [31] for the non-collagenated (DBBM) and collagenated xenografts, respectively. The ARRIVE guidelines and SYRCLE risk of bias tool for animal studies were adopted.

### 2.2. Power Calculation and Sample Size

A one-tail analysis of the raw data on the total bone formed within the elevated space from the two experiments [30,31] was performed, with a power of 0.84, α = 0.05, and an effect size of 1.68. It was assumed that different proportions of new bone could be found between the two groups in the various regions evaluated. A sample size of six animals was considered sufficient to reject the null hypothesis that the population means of the experimental and control groups were equal. G*Power 3.1 was used for the calculations.

### 2.3. Experimental Design

The histological slides from two experiments, in which two different biomaterials were used as grafts for sinus floor elevation, were further evaluated to analyze other variables and different regions. Other data have been reported elsewhere [30,31]. In both studies, 18 male New Zealand rabbits (3.5–4.0 kg with a mean age of 4–5 months) were used. The maxillary sinuses of rabbits were elevated using xenografts with different features. In one study [30], a non-collagenated deproteinized bovine bone mineral (DBBM) with either 0.250–1.0 mm or 1.0–2.0 mm granules (Bio-Oss^®^, Geistlich Biomaterials, Wolhusen, LU, Switzerland) was used as filler material, while in the other study [31], a collagenated corticocancellous porcine bone (OsteoBiol Gen-Os, 0.250–1.0 μm, Tecnoss, Giaveno, Italy) was applied. Both studies were randomized.

Randomization was performed electronically by an operator not involved in the surgeries and was maintained in sealed opaque envelopes that were opened after exposure of the nasal bone [31] or after elevation of the sinus mucosa bilaterally [30].

In each study, three groups were formed; each was composed of six animals that were euthanized after 2, 4, and 8 weeks.

### 2.4. Xenografts 

Bio-Oss (Geistlich Biomaterials, Wolhusen, LU, Switzerland) is a bovine bone deproteinized using a strong alkalis and organic solvent treatment at a temperature of 300 °C. The total porosity, as evaluated by the mercury intrusion method, was 63.5%, whereas the intraparticle porosity was 51%. The real density was 3.21 g·cm^−3^ and the mineral content was 95% [32].

Osteobiol Gen-Os (Tecnoss, Giaveno, Italy) is a porcine bone treated at a low temperature of up to 130 °C to eliminate the pathogens that allow the preservation of the structure and composition of both collagen and hydroxyapatite. The total porosity, as evaluated by the mercury intrusion method, was 33.1%, whereas the intraparticle porosity was 21%. The real density was 2.43 g·cm^−3^ and the mineral content was 64.6% [32].

### 2.5. Surgical Procedures

Acepromazine (1.0 mg/kg; Acepran^®^, Vetnil, Louveira, São Paulo, Brazil) was administered subcutaneously to induce anesthesia, followed by intramuscular (IM) injection of a mixture of xilazine (3.0 mg/kg; Dopaser^®^, Hertape Calier, Juatuba, Minas Gerais, Brazil) and 60 mg/kg ketamine (50.0 mg/kg; União Química Farmacêutica Nacional S/A, Embuguaçú, São Paulo, Brazil). Local anesthesia was injected into the experimental zone.

In the DBBM group, the skin, muscle, and periosteum on the nasal dorsum were incised and the nasal bone was exposed. Access windows were prepared bilaterally with drills, the sinus mucosa was elevated with small curettes, and the DBBM, either small or large granules, was placed inside the elevated spaces. The access windows were covered with a collagen membrane (Bio-Gide^®^ Geistlich Biomaterials, Wolhusen, LU, Switzerland), and the overlying soft tissues were subsequently sutured. Similar surgical procedures were performed for the collagenated group. However, the access windows were prepared using either drills or an air sonic instrument (Sonosurgery, TeKne Dental, Calenzano, Florence, Italy) and micro-saw (SFS 102, Komet-Brasseler-GmbH, Rastatt, Germany). The elevated space was filled with collagenated corticocancellous porcine bone. The access windows were covered using a collagen membrane (Evolution, 0.3 mm, OsteoBiol Gen-Os; Tecnoss, Giaveno, Italy) and the soft tissues were closed with internal resorbable sutures (Polyglactin 910 5-0, Vicryl^®^, Ethicon, Johnson & Johnson, São José dos Campos, Brazil) and the skin was sutured with nylon (Ethilon 4-0^®^, Ethicon, Johnson & Johnson, São José dos Campos, Brazil).

Three healing periods were analyzed after 2, 4, and 8 weeks. In both studies, six animals were used for each healing period.

### 2.6. Maintenance Care

The animals were maintained in an acclimatized room at the animal facilities of the Faculty of Dentistry in Ribeirão Preto, University of São Paulo. The animals were housed in individual cages with ad libitum access to food and water. The monitoring of wounds and biological functions was performed daily by expert personnel. In the post-operative period, ketoprofen twice a day for 2 days (3.0 mg/kg, Ketofen 10%, Merial, Campinas, São Paulo, Brazil) was injected IM while Tramadol 2% twice a day for 2 days (1.0 mg/kg, Cronidor, Agener União Saúde Animal, Apucarana, Paraná, Brazil) was injected subcutaneously.

### 2.7. Euthanasia

Animals were anesthetized in a manner similar to that of the surgical session. Subsequently, an overdose of sodium thiopental (1.0 g, 2 mL, Thiopentax^®^, Cristália Produtos Químicos Farmacêuticos, Itapira, São Paulo, Brazil) was used to euthanize the animals.

### 2.8. Histological Preparation

The biopsies were placed in 10% buffered formalin and scanned for micro-CT analysis. The samples were dehydrated in increasing concentrations of ethanol (50% to 100%) for 6 days and subsequently included in an ascending series of resin solutions (50% to 100%; LR White^TM^ hard grid, London Resin Co Ltd., Berkshire, UK). Thermal polymerization was performed in an oven at 60 °C for 24 h. After polymerization, the biopsies were cut in a coronal plane, crossing the small screw placed between the two sinuses, using high precision cutting equipment (Exakt, Apparatebau, Norderstedt, Germany). Two hemi-biopsies were obtained from which two slides of approximately 150 µm were prepared. Using a sequence of sandpapers of decreasing grain size mounted on a grinding machine (Exakt, Apparatebau, Norderstedt, Germany), the two slices were ground to a thickness of approximately 60 µm and polished. Two sections were obtained from each sample and stained with toluidine blue or Stevenel’s blue and alizarin red.

### 2.9. Data Analysis

The surgery in both groups was performed by the same surgeon (E.R.S.; see acknowledgements), and both histological measurements were performed by a well-trained examiner (K.A.A.A.; see acknowledgements). Before starting the measurements, a calibration with another professional (D.B.; see acknowledgements) was performed, and it was carried out until the inter-rater agreement in the identification of the tissues achieved a Cohen’s coefficient k > 0.80.

To maintain high similarity and homogeneity, only one sinus was used in each experiment in the present study, that is, with the small granules in the DBBM experiment (DBBM group) and with the access window prepared with drills in the collagenated corticocancellous porcine bone experiment (collagenated group).

Histological measurements were performed using NIS-Elements D software (v 4.0, Laboratory Imaging, Nikon Corporation, Tokyo, Japan) on an Eclipse Ci microscope (Nikon Corporation, Tokyo, Japan) equipped with a video camera (Digital Sight DS-2Mv, Nikon Corporation, Tokyo, Japan). A point-counting procedure was used, superposing a lattice with squares of 75 µm in dimension on the image of the histological slide at ×100 magnification.

The following tissues were evaluated in both studies: mineralized new bone, xenografts, inflammatory infiltrate areas, vessels, osteoclastic zones, and soft tissues (marrow spaces, connective tissues, and matrix tissue). The tissues were evaluated in three different regions: close to the sinus bone walls (bone walls region), subjacent sinus mucosa (submucosa region), and in the center of the elevated space (middle region).

Two groups were defined: the DBBM and collagenated groups. The primary variables were the percentage of new bone and xenograft remnants. All the other parameters were secondary variables. Mean values and standard deviations were calculated for each variable using the software Excel 2013 (Microsoft Corporation, Redmond, WA, USA). 

Statistical analyses were performed for both primary and secondary variables using the Prism software (Version 9.4.1, IBM Inc., Chicago, IL, USA). The Shapiro-Wilk test for the assessment of normality was applied for all variables. The unpaired *t*-test or Mann-Whitney test was used for statistical analyses. The significance level was set at *p* < 0.05.

## 3. Results

All periods are presented as n = 6, and data were available for all specimens. Data from three regions were evaluated (Figure 1A–D): bone walls (Table 1), sub-mucosa (Table 2), and middle (Table 3) regions.

In the DBBM group, the area of the elevated space decreased by 5.6% between 2 and 4 weeks (*p* = 0.599) and by 16.3% between 2 and 8 weeks (*p* = 0.108). In the collagenated group, the reduction in area was 33.7% and 52.2% between 2 and 4 weeks (*p* = 0.001) and between 2 and 8 weeks (*p* < 0.001), respectively.

After 2 weeks of healing, new bone was observed in both groups, especially close to the bone walls (Figure 2A,B).

Higher bone formation was observed in the DBBM group than in the collagen group during the healing period (*p* = 0.011). After 8 weeks of healing, new bone increased over time in all regions in both groups, reaching the highest value in the collagenated group. In this period of healing, the only statistically significant difference in bone formation was found in the bone walls region (*p* = 0.004), with a fraction of 28.6 ± 5.3% in the DDBM group and 41.6 ± 6.5% in the collagenated group (Figure 3).

The xenograft content in the DBBM group decreased by ~6% between weeks 2 and 8 in the bone wall region (*p* = 0.064), ~3% in the submucosal region (*p* = 0.598), and ~11.5% in the middle region (*p* = 0.029). The collagenated group presented much higher resorption, reducing the percentages from about 43–53% to 5–6% in the same interval of healing. The differences were statistically significant in all regions after 4 and 8 weeks of healing (*p* < 0.01; Figure 4).

After 8 weeks of healing, higher soft tissue content was present in the collagenated group than in the DBBM group, and the difference was statistically significant in all regions (*p* < 0.001; Figure 5A,B).

The osteoclastic zones were limited in extension after 2 weeks of healing. A higher concentration of these zones was observed in the bone wall region of the collagenated group (Figure 6). 

After 4 weeks, the osteoclastic zones remained stable in the DBBM group and increased in the collagenated group, reaching 9.6% in the sub-mucosa region (Figure 7A,B).

In this period, the differences between the groups were statistically significant in all regions (*p* < 0.05). The osteoclastic zone content decreased after 8 weeks to values similar to or lower than those in the 2-week period.

After 2 weeks of healing, the range of the percentages of vessels was 2.1–4.2% in the DBBM group, and 2.3–2.6% in the collagenated group (Figure 8). In the DBBM group, after 8 weeks of healing, the vessel content was <1% in all regions. Conversely, the proportion of vessels in the collagenated group increased over time, reaching fractions 9–10% after 8 weeks (*p* < 0.01). In this period, the differences between the groups were statistically significant for all the regions evaluated.

After 2 weeks of healing, the inflammatory infiltrate was low (<1%) in all regions. The percentages remained low (≤0.2%) in the DBBM group, increased in the collagenated group after 4 weeks, and ranged between 0.7–1.6% after 8 weeks.

## 4. Discussion

The dimensions of the elevated space decreased over time in both the DBBM and collagenated groups. However, between 2 and 8 weeks of healing, a greater reduction in the dimensions was observed in the collagenated group, with >50% of the total area, whereas in the DBBM group, the reduction was ~16%. In the DBBM group, dimensional variations were similar to those reported for humans [9]. In contrast, in the collagenated group, the shrinkage was higher than that reported in a clinical study in which the dimensional variations evaluated on CBCTs 9 months after sinus lifting ranged between 18% and 34% [33,34]. Nevertheless, it should be considered that in other clinical situations, such as ridge preservation after tooth extraction, no differences between collagenated and DBBM were found regarding the maintenance of soft tissue dimensions [35].

In the present study, after 2 weeks, new bone formation was observed in the sinus bone walls in both groups. This is in agreement with several other experimental studies that showed a similar amount of healing [36,37,38,39,40]. In an experiment on minipigs, sinus augmentation was performed using DBBM or an aqueous paste of synthetic nanoparticulated hydroxyapatite [40]. Healing was evaluated after 6 and 12 weeks in three regions that were progressively more distant from the sinus bone walls. The farther from the bone walls, the lower the bone volume observed, confirming that bone was formed from the sinus bone walls. In the present study, in the first period of healing, higher proportions of new bone were found closer to the bone walls than to the others. The influence of bone walls on bone formation was also shown in a clinical study [41]. Biopsies were collected from 18 patients 6 months after sinus floor elevation. A higher percentage of new bone was observed in the mesial sites, closer to the mesial bone wall, than in the distal sites. Moreover, a strong negative correlation between bone percentage and sinus width was found. This, in turn, means that the greater the distance between the lateral and palatal walls, the lower is bone formation. 

In the sub-mucosa region, new bone reached percentages similar to the other regions only in the last period assessed, that is, after 8 weeks of healing. This might be because the sinus mucosa did not participate in bone formation in the earliest period of healing, despite its potential to form bone [42,43]. This finding is in agreement with the data from other experimental studies [8,17,36,37,44]. After 8 weeks, higher percentages of new bone were found in the collagenated group than in the DBBM group in the bone wall region. The percentages of new bone found after 8 weeks of healing in the present study were similar to those reported in human histological studies. In a clinical study [45], 48 sinuses were elevated with either DBBM or synthetic graft. Biopsies were collected 6–8 months after surgery. The percentages of mineralized bone were 19.8% and 21.6%, respectively. In another clinical study [46], bovine cancellous bone obtained at a high temperature (>1200 °C) was used for sinus floor elevation. Mini-implants were installed 6 months after healing and were retrieved after another 3 months. Histological examination of the biopsies revealed 21–23% mineralized bone. In sinuses elevated with a collagenated xenograft similar to that used in the present study, and with mini-implants again installed after 6 months of healing and retrieved after 3 months, percentages of mineralized bone ranging between ~32–39% were found [47,48,49]. In another clinical study [50], biopsies collected through a crestal approach revealed 40.1% mineralized bone. The percentages of mineralized bone reported in the above-mentioned studies were similar to those found close to the bone walls in the present study. This outcome might be related to the region from which the biopsies were collected in clinical studies, that is, the bone crest. This region contained the residual bone crest, and it was close to the floor and sinus walls, a condition that resembles that in the experiment in rabbits.

This aspect has important clinical relevance, because the region close to the base of the sinus is where the implants will be installed. Therefore, efforts should be made to preserve this region, for example, by making the access windows as small as possible and far away from the sinus floor to increase the source of new bone from the lateral sinus wall [33,34].

The higher percentages of new bone found in the collagenated groups compared to the DBBM group in the bone wall region matches the lower content of xenograft residual in the collagenated group compared to the DBBM group; in fact, a higher resorption of biomaterial was observed in the former than in the latter group. After 8 weeks of healing, >40% of DBBM and <6% of collagenated corticocancellous porcine bone were still present in all regions. This outcome is in agreement with other studies that showed a higher maintenance of DBBM granules over time compared to collagenated xenografts [26,30]. It must be considered that the presence of the biomaterial inside the bone (composite bone) might interfere with the osseointegration of implants installed both in a simultaneous [51] or delayed fashion [46,47,48,49]. In a rabbit experiment [51], implants were placed simultaneously with sinus augmentation performed with a non-collagenated xenograft. Granules of the biomaterial were found in contact with the implant surface in percentages ranging between 1.7% and 3.6%. Biopsies of mini-implants installed 6 months after sinus floor elevation were collected from humans after a further 3 months of healing [46,47,48,49]. The implant surface was found to be in contact with granules of biomaterials that hampered the growth of newly formed bone on the implant surface. The percentages presented a large range when a collagenated xenograft was used (0.6 to 15.9%) [47,48,49] but reached 25.1% to 27.2% when a non-collagenated xenograft was used [46].

Higher amounts of soft tissue were found in the collagenated group than in the DBBM group, with a statistically significant difference after 8 weeks in all regions. This tissue was mainly composed of marrow spaces in the bone wall region and, in the middle and submucosa regions, by provisional matrix that was not yet completely involved in the remodeling process (Figure 5A, B).

In the present study, the percentage of osteoclastic zones was higher in the collagenated group than in the DBBM group, which is in accordance with the different levels of resorption of the biomaterial in the two groups. The highest percentages of resorptive zones were observed after 4 weeks of healing in the collagenated group, especially in the sub-mucosa region. Considering the tendency of the sinus to regain its original size [6,7,8,28,33,34], the higher rate of graft resorption in the collagenated group than in the DBBM group contributed to the greater loss of dimension of the elevated space observed in the former than in the latter. 

In the DBBM group, the proportion of vessels decreased between 2 and 8 weeks, while in the collagenated group, the content in vessels continuously increased, which might influence healing. Blood provides nutrients, gas exchange, immune system cells, mesenchymal stem cells, and growth factors [52,53,54]. It has been shown that various biomaterials, including xenogeneic grafts, were associated pro-angiogenic effect and an increased expression of various grow factors [53].

The higher content of inflammatory infiltrates, although limited in dimension, in the collagenated group than in the DBBM group was related to higher xenograft resorption in the former than in the latter. The present study did not include gene expression analysis. Nevertheless, it should be considered that lymphocytes play an important role in the immune system and determine the specificity of the immune response following infections [55]. Damage to the bone triggers the immune response and the activation of neutrophils, mast cells, monocytes, and macrophages [56]. It has been shown in mice that the absence of lymphocytes B and T compromises bone regeneration in fractures [57]. In an experiment in rabbit femurs [58], gene expression analysis was performed around implants and in sham sites. Upregulation of the immune system and downregulation of bone resorption markers were observed at the implant sites compared with the sham sites.

The present study also demonstrated differences in bone formation and graft resorption between collagenated and non-collagenated xenografts. DBBM presented a very low rate of resorption, and the bone was formed in close contact with the granules of the graft that were interconnected by a bridge of newly formed bone. Bone formation in rabbits was similar to that described in a previous study [26]. In the early stages of healing, the DBBM granules were surrounded by dense tissue, whereas loose tissue filled the spaces among the granules. Over time, the dense tissue was replaced by new bone in contact with the graft surface, whereas the loose tissue was replaced by primitive marrow spaces. The collagenated graft showed mixed characteristics, including bone formation around graft granules, similar to DBBM, and sites with several osteoclastic zones engendering an extensive graft resorption similar to that shown for autogenous bone [59].

The surgical procedures performed in the experiments included in the present study were conducted under sterile conditions, although such conditions could not be completely obtained in the oral environment. Nevertheless, it might help to reduce the possible contamination of instruments, biomaterials, and the operative field [60,61]. Technologies and procedures may be implemented with the aim of better sterilizing the operating field and improving healing [62].

Further studies that include longer healing periods and the evaluation of different biomaterials may be performed. Synthetic biomaterials should be studied further. In fact, the increased demand for bone regeneration, as well as the morbidity of the donor sites for autogenous bone, the possible presence of prions within xenogenous grafts, and patients’ religious beliefs have increased interest in studying synthetic (alloplastic) materials. Among them, biphasic β-tricalcium phosphate/hydroxyapatite (β-TCP/HA) and bioactive glasses presented interesting results. It was recently showed that at β-TCP/HA graft new bone was not only surrounding the particles but also spreading inward them forming a structure called by the authors “interpenetrating bone network” [63]. Bioactive glasses have also been widely used in maxillofacial surgery, providing interesting results in bone formation, even better than those obtained with other synthetic biomaterials [64].

In the present study, two experiments were selected for homogeneity in methodology. The experiments were performed in the same laboratory by the same research group. The healing periods were similar, and histological slides were prepared in the same laboratory using the same methods and analyzed by the same expert assessor. To increase homogeneity, only one sinus was used for each experiment, that is, with the small granules in the DBBM experiment (DBBM group) and that with the access window prepared with drills in the collagenated porcine xenograft experiment (collagenated group). In turn, this means that both access windows were prepared using drills. 

Despite efforts to increase the homogeneity of the methodology, various limitations must be mentioned. The dimensions of the two access windows were slightly different: 3.5 mm × 6 mm (21 mm^2^) in the non-collagenated group and 4 mm × 4 mm (16 mm^2^) in the DBBM group. However, access windows of 18 mm^2^ and 30 mm^2^ did not show differences in bone formation in a similar study in rabbits [65]. The different collagen membranes that covered the osteotomies in the two experiments might be considered another limitation of the study. However, it is unlikely that this difference affected the healing of the internal sides of the elevated spaces. This has been substantiated by both animal [66] and human [46] studies. The model used should also be considered as a limitation. In fact, when interpreting the results of the present study to humans, the different speeds of healing are an important topic to be considered [67]. Other important limitations are the short period of evaluation and the selection of material from previous studies, even though efforts were made to maintain a high homogeneity between the two studies.

## 5. Conclusions

The present study showed that both xenografts allowed new bone formation. In comparison with a non-collagenated xenograft, a collagenated xenograft underwent higher resorption, resulting in greater shrinkage of the elevated space after sinus lifting and a higher content of new bone in the regions close to the bone walls. However, considering the limitations of the present study, the results should be interpreted with caution.

## Figures and Tables

**Figure 1 jfb-13-00276-f001:**
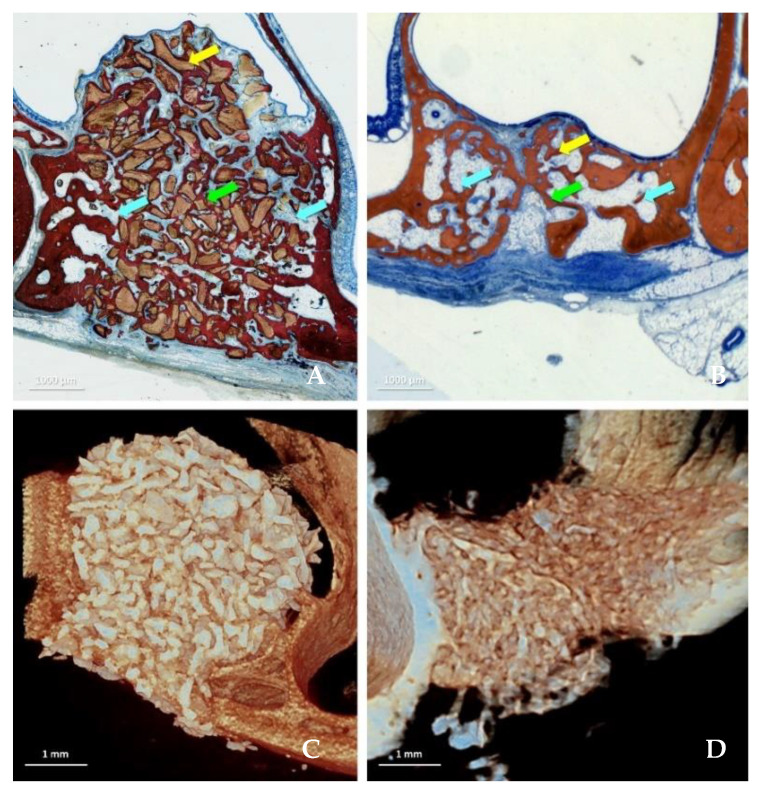
Healing after 8 weeks in the DBBM (**A**,**C**) and collagenated (**B**,**D**) groups illustrated in histological (**A**,**B**) and microCT images (**C**,**D**). The tissues were evaluated in three different regions: close to the sinus bone walls (bone walls region, light blue arrows), subjacent the sinus mucosa (sub-mucosa region, yellow arrow) and in the center of the elevated space (middle region, green arrow). Stevenel’s blue and alizarin red stain.

**Figure 2 jfb-13-00276-f002:**
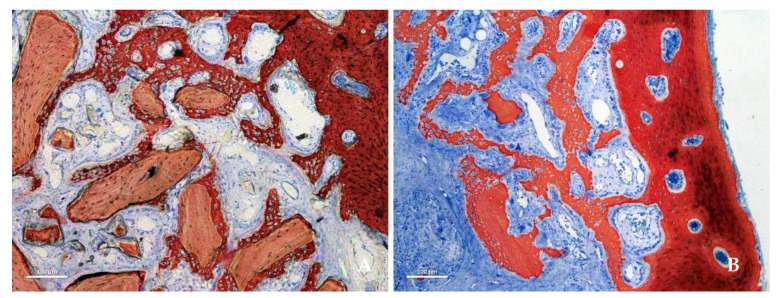
Photomicrographs of ground sections representing the healing after 2 weeks. New bone was formed from the sinus bone walls and surrounded the neighbor granules. (**A**) DBBM group. (**B**) collagenated group. Stevenel’s blue and alizarin red stain.

**Figure 3 jfb-13-00276-f003:**
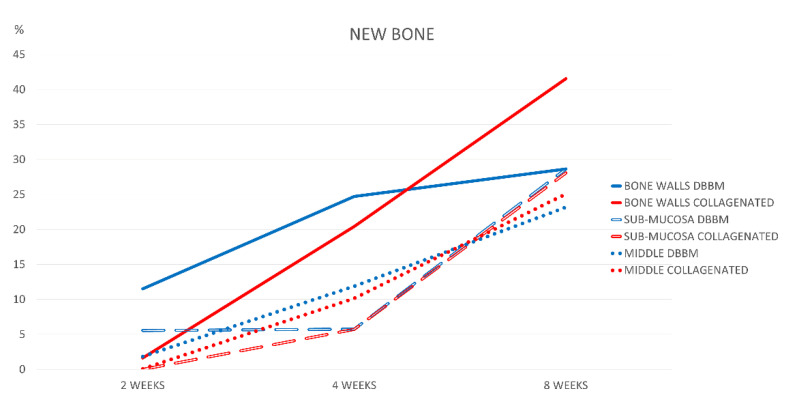
Graph illustrating percentage bone formation in the various regions examined after 2, 4, and 8 weeks of healing.

**Figure 4 jfb-13-00276-f004:**
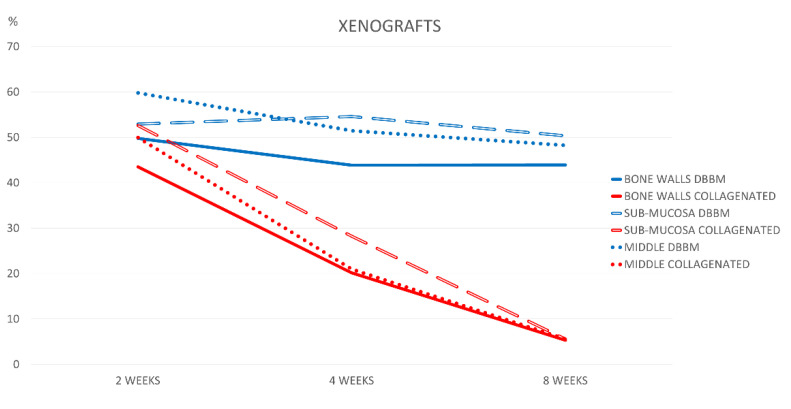
Graph illustrating percentage xenograft resorption in the various regions examined after 2, 4, and 8 weeks of healing.

**Figure 5 jfb-13-00276-f005:**
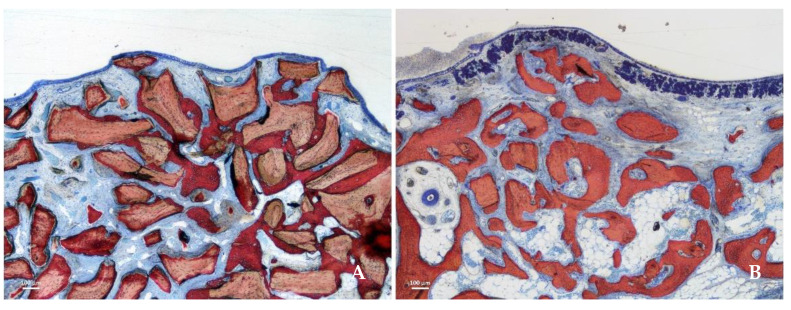
Photomicrographs of ground sections representing the healing after 8 weeks in the submucosa region. Xenograft particles and soft tissues were present in both groups, but in higher percentages in the DBBM group. (**A**) DBBM group. (**B**) Collagenated group. Stevenel’s blue and alizarin red stain.

**Figure 6 jfb-13-00276-f006:**
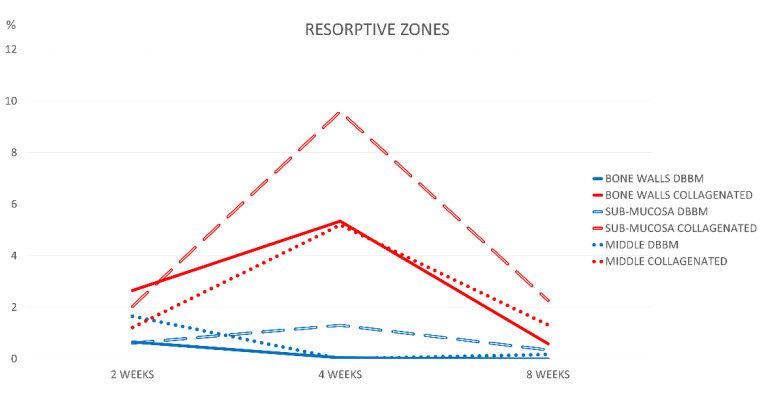
Graph illustrating the percentages of resorptive zones in the various regions examined after 2, 4, and 8 weeks of healing.

**Figure 7 jfb-13-00276-f007:**
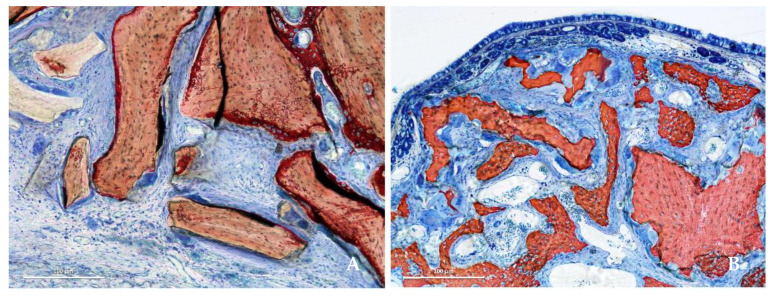
Photomicrographs of ground sections showing resorptive zones after 4 weeks. (**A**) DBBM group. (**B**) Collagenated group. Stevenel’s blue and alizarin red stain.

**Figure 8 jfb-13-00276-f008:**
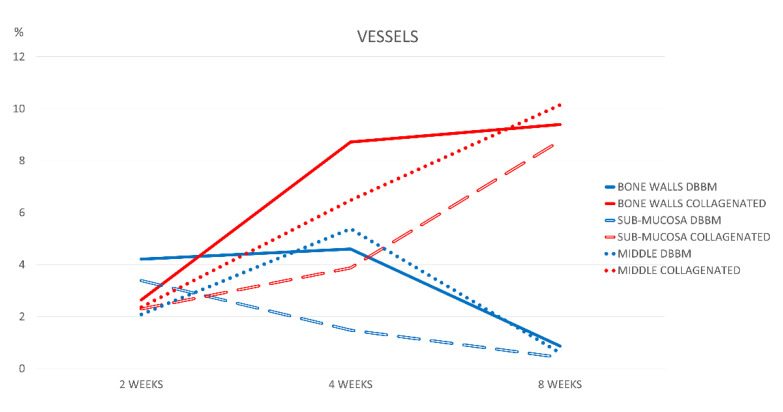
Graph illustrating the percentages of vessels in the various regions examined after 2, 4, and 8 weeks of healing.

**Table 1 jfb-13-00276-t001:** Tissues components in the bone walls region in the DBBM and collagenated groups at the various periods of healing. Mean values ± standard deviation in percentages.

	New Bone	Xenograft	Soft Tissues	Osteoclastic Zones	Vessels	Inflammatory Infiltrate
2 Weeks DBBM	11.5 ± 7.6 *	49.8 ± 5.3 *	33.9 ± 7.5 *	0.7 ± 0.9 *	4.2 ± 3.8	0.0 ± 0.0 *
2 Weeks Collagenated	1.6 ± 1.7 *	43.5 ± 4.3 *	48.8 ± 3.5 *	2.7 ± 0.8 *	2.6 ± 1.5	0.7 ± 0.6 *
*p* value	0.011	0.048	0.001	0.010	0.810	0.002
4 Weeks DBBM	24.7 ± 6.2	43.9 ± 4.1 *	26.8 ± 6.6 *	0.0 ± 0.1 *	4.6 ± 1.3	0.0 ± 0.0 *
4 Weeks Collagenated	20.4 ± 15.2	20.2 ± 12.9 *	44.4 ± 7.3 *	5.3 ± 2.6 *	8.7 ± 5.0	0.9 ± 1.0 *
*p* value	0.535	0.002	0.001	0.003	0.079	0.007
8 Weeks DBBM	28.6 ± 5.3 *	43.9 ± 4.4 *	26.3 ± 2.5 *	0.0 ± 0.0 *	0.9 ± 0.7 *	0.2 ± 0.5
8 Weeks Collagenated	41.6 ± 6.5 *	5.3 ± 4.3 *	43.1 ± 5.5 *	0.6 ± 0.5 *	9.4 ± 3.1 *	0.1 ± 0.3
*p* value	0.004	0.004	0.004	0.002	<0.0001	0.902

* *p* < 0.05 between deproteinized bovine bone mineral (DBBM) and collagenated corticocancellous porcine bone (collagenated) groups.

**Table 2 jfb-13-00276-t002:** Tissues components in the sub-mucosa region in the DBBM and collagenated groups at the various periods of healing. Mean values ± standard deviation in percentages.

	New Bone	Xenograft	Soft Tissues	Osteoclastic Zones	Vessels	Inflammatory Infiltrate
2 Weeks DBBM	5.6 ± 13.6	53.0 ± 10.7	37.5 ± 5.5	0.6 ± 0.8 *	3.4 ± 3.1	0.0 ± 0.0
2 Weeks Collagenated	0.0 ± 0.0	52.6 ± 5.5	42.8 ± 4.6	2.0 ± 1.0 *	2.3 ± 1.4	0.3 ± 0.4
*p* value	0.317	0.631	0.100	0.024	0.457	0.059
4 Weeks DBBM	5.7 ± 7.8	54.6 ± 9.4 *	36.9 ± 8.9 *	1.3 ± 1.4 *	1.5 ± 1.3	0.0 ± 0.0 *
4 Weeks Collagenated	5.7 ± 11.2	28.2 ± 1.8 *	50.1 ± 10.7 *	9.6 ± 7.5 *	3.9 ± 3.9	2.5 ± 2.2 *
*p* value	0.494	0.002	0.042	0.024	0.186	0.007
8 Weeks DBBM	28.6 ± 14.0	50.3 ± 4.6 *	20.2 ± 10.4 *	0.3 ± 0.9	0.4 ± 1.1 *	0.0 ± 0.0 *
8 Weeks Collagenated	28.1 ± 16.4	5.6 ± 4.4 *	54.6 ± 21.5 *	2.3 ± 3.4	8.8 ± 5.0 *	0.7 ± 0.6 *
*p* value	0.950	<0.0001	0.006	0.216	0.003	0.022

* *p* < 0.05 between deproteinized bovine bone mineral (DBBM) and collagenated corticocancellous porcine bone (collagenated) groups.

**Table 3 jfb-13-00276-t003:** Tissues components in the middle region in the DBBM and collagenated groups at the various periods of healing. Mean values ± standard deviation in percentages.

	New Bone	Xenograft	Soft Tissues	Osteoclastic Zones	Vessels	Inflammatory Infiltrate
2 Weeks DBBM	1.8 ± 4.5	59.8 ± 8.6 *	34.3 ± 6.9 *	1.6 ± 1.6	2.1 ± 1.8	0.3 ± 0.9
2 Weeks Collagenated	0.1 ± 0.2	50.0 ± 6.3 *	46.0 ± 6.5 *	1.2 ± 0.5	2.4 ± 1.1	0.4 ± 0.4
*p* value	0.902	0.048	0.032	0.571	0.470	0.210
4 Weeks DBBM	11.9 ± 6.0	51.5 ± 14.9 *	31.3 ± 7.5 *	0.0 ± 0.0 *	5.4 ± 4.1	0.0 ± 0.0 *
4 Weeks Collagenated	10.2 ± 18.2	21.0 ± 19.7 *	55.8 ± 16.2 *	5.2 ± 2.8 *	6.5 ± 5.9	1.4 ± 1.4 *
*p* value	0.149	0.013	0.007	0.002	0.719	0.022
8 Weeks DBBM	23.2 ± 12.8	48.3 ± 7.0 *	27.8 ± 7.0 *	0.2 ± 0.4	0.6 ± 0.8 *	0.0 ± 0.0 *
8 Weeks Collagenated	25.1 ± 8.0	5.6 ± 3.2 *	56.2 ± 12.8 *	1.3 ± 2.5	10.1 ± 4.9 *	1.6 ± 2.0 *
*p* value	0.764	<0.0001	0.001	0.216	0.006	0.022

* *p* < 0.05 between deproteinized bovine bone mineral (DBBM) and collagenated corticocancellous porcine bone (collagenated) groups.

## Data Availability

The data are available on reasonable request.

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
