# Peer review of "Healing Patterns of Non-Collagenated Bovine and Collagenated Porcine Xenografts Used for Sinus Floor Elevation: A Histological Study in Rabbits"

_jfb, 2022, doi:10.3390/jfb13040276_

Round 1

Reviewer 1 Report

“Healing patterns of non-collagenated bovine and collagenated porcine xenografts used for sinus floor elevation. A histological study in rabbits.” was submitted to JFB.

This study aimed to compare the healing of collagenated and not collagenated xenografts used for maxillary sinus floor elevation. The authors concluded that both xenografts allowed new bone formation; however, in comparison with a non-collagenated xenograft, a collagenated xenograft underwent higher resorption.

The manuscript deals with an interesting issue; however, several concerns are related to the study.

Abstract

-. Objective: The outcome variables that will allow this comparison to be established must be defined.

-. Results: lines 23 and 25. Please present P-values. Were there statistically significant differences?

-. Lines 26-27. Present the value of the amount and if there was a statistically significant difference.

Keywords: animal study, sinus floor elevation, and collagen membrane are not MeSH terms.

Introduction

-. Page 2, lines 57,58. Please present the advantages and disadvantages of these two bone substitutes.

-. Page 2, line 62. Describe the healing characteristics and resorption rates.

-. Page 2. Line 64. Be specific about missing data.

-. Page 2. Line 65. Clearly explain the reason for selecting these two biomaterials.

-. Page 2. Line 66. Present the outcome variables that will allow such a comparison.

Methods

-. Page 2, line 95. Present in detail how the randomization was carried out.

-. Surgical procedures. Lines 121-122. The implications of this difference should be presented in the discussion.

-. The two surgical protocols present differences. Its implications should also be discussed.

-. Did the assessor who performed the histological measurements have any calibration training? If so, present the results and the statistical test used.

-. Please indicate the statistical test used to establish the normal distribution of the data, and its result.

Results

-. Page 6, lines 197-202. Please present P-values.

-. Page 7, lines 204-207. For clarity present P-values.

-. Page 8, lines 218-220. Please present P-values.

-. Page 8, lines 221-222. Please present P-values.

-. Page 9, lines 233-235. Please present P-values.

Discussion.

-. Page 9, lines 241-243. An explanation for that difference should be provided.

-. Page 10, lines 278-280. Present the bases for an increase in blood vessels.

-. The results should be contrasted with more studies on humans.

-. The study has more limitations that should be mentioned.

Conclusions

They should be presented indicating that the study has limitations, and the results should be evaluated with caution.

The language must be edited.

Author Response

Open Review

English language and style

( ) English very difficult to understand/incomprehensible
( ) Extensive editing of English language and style required
(x) Moderate English changes required
( ) English language and style are fine/minor spell check required
( ) I don't feel qualified to judge about the English language and style

Yes

Can be improved

Must be improved

Not applicable

Does the introduction provide sufficient background and include all relevant references?

( )

( )

(x)

( )

Are all the cited references relevant to the research?

( )

(x)

( )

( )

Is the research design appropriate?

(x)

( )

( )

( )

Are the methods adequately described?

( )

(x)

( )

( )

Are the results clearly presented?

( )

(x)

( )

( )

Are the conclusions supported by the results?

( )

(x)

( )

( )

Comments and Suggestions for Authors

“Healing patterns of non-collagenated bovine and collagenated porcine xenografts used for sinus floor elevation. A histological study in rabbits.” was submitted to JFB.

This study aimed to compare the healing of collagenated and not collagenated xenografts used for maxillary sinus floor elevation. The authors concluded that both xenografts allowed new bone formation; however, in comparison with a non-collagenated xenograft, a collagenated xenograft underwent higher resorption.

The manuscript deals with an interesting issue; however, several concerns are related to the study.

Authors’ answer.

Dear reviewer, thank you for your valuable comments aimed at increasing the quality of this article. We hope that we have fulfilled your recommendations. Kind regards.

The authors

Abstract

-. Objective: The outcome variables that will allow this comparison to be established must be defined.

Authors’ answer.

We have added the following phrase (line 22): “The primary variables were new bone and xenograft remnants percentages.”

-. Results: lines 23 and 25. Please present P-values. Were there statistically significant differences?

Authors’ answer.

We have added both p values as follows (lines 24-27): “(p<0.01 between the two areas after 8 weeks)” and rephrased “(41.6% and 28.6%, respectively; p<0.01).

-. Lines 26-27. Present the value of the amount and if there was a statistically significant difference.

Authors’ answer.

We have rephrased the sentence as follows (lines 28-29): “A higher percentage of vessels in all regions evaluated (p<0.01) and soft tissue in the sub-mucosa region (p<0.05) was found in the collagenated group than in the DBBM group.”

Keywords: animal study, sinus floor elevation, and collagen membrane are not MeSH terms.

Authors’ answer.

We have updated the key words as follows:

“Bone Substitutes; Sinus Floor Augmentation; Bone Formation; Bone Regeneration; Bone Resorption; Animal Model”

Introduction

-. Page 2, lines 57,58. Please present the advantages and disadvantages of these two bone substitutes.

Authors’ answer.

We have rephrased this part as follows (lines 62-68): “A systematic review with meta-analysis [12] concluded that autogenous bone was the best choice when a high content of newly formed bone was needed. Furthermore, other bio-materials might be considered reliable bone substitutes in cases of concern regarding the donor site. The outcomes from the review showed significantly greater new bone formation with bovine bone than with hydroxyapatite alone, and better results with a biphasic graft composed of β-tricalcium phosphate and hydroxyapatite than with bovine bone.”

-. Page 2, line 62. Describe the healing characteristics and resorption rates.

Authors’ answer.

 We have added the following part (lines 71-88):

“Different healing features and resorption rates of different biomaterials have also been reported in animal experiments [26-29]. In a rabbit model [26], sinus augmentation was performed either with deproteinized bovine bone mineral (DBBM) or with a collagen sponge. Healing was evaluated after different periods and showed stable dimensions of the elevated space between 7 and 40 days, while at the collagen site, >50% of the dimensions were lost. The total amount of new bone was higher at DBBM sites than at collagen sponge sites.

In a similar study in rabbits [27], sinus augmentation was performed using either DBBM or autogenous bone, and simultaneous implants were installed. Healing was evaluated after 7 and 40 days. After 40 days of healing, a higher percentage of new bone was found at autogenous sites than at DBBM sites. However, 2/3 of the area of the elevated space was lost at the autogenous sites, while the dimensions were maintained at the DBBM sites.

In another similar study [28], sinus floor augmentation was performed bilaterally using collagenated corticocancellous porcine bones. At the test site, a collagen membrane was placed adjacent to the sinus mucosa prior to grafting. Similar amounts of new bone were found in both the sinuses. However, approximately 50% of the elevated space was lost during healing.”

-. Page 2. Line 64. Be specific about missing data.

-. Page 2. Line 65. Clearly explain the reason for selecting these two biomaterials.

-. Page 2. Line 66. Present the outcome variables that will allow such a comparison.

Authors’ answers to the three comments.

We tried to implement this part as follows (lines 89-96):

“These studies showed similar percentages of new bone [26,28] but a higher loss in the dimension of the elevated space in the collagenated group than in the non-collagenated group. A comparison between these two different xenografts on different parameters, such as bone formation, graft resorption, vessel formation, presence of resorptive zones, and inflammatory infiltrates, might still provide useful information. Moreover, detailed data on these parameters in different regions of the elevated regions, such as those close to the bone walls or sinus mucosa, might allow the disclosure of differences in healing between the two biomaterials.”

Methods

-. Page 2, line 95. Present in detail how the randomization was carried out.

Authors’ answer.

We have rephrased the sentence as follows (lines 127-129):

“Randomization was performed electronically by an operator not involved in the surgeries and was maintained in sealed opaque envelopes that were opened after exposure of the nasal bone [31] or after elevation of the sinus mucosa bilaterally [30].

-. Surgical procedures. Lines 121-122. The implications of this difference should be presented in the discussion.

-. The two surgical protocols present differences. Its implications should also be discussed.

Authors’ answers to both comments.

We have rephrased one part as follows (lines 407-411): “To increase homogeneity, only one sinus was used for each experiment, that is, with the small granules in the DBBM experiment (DBBM group) and that with the access window prepared with drills in the collagenated porcine xenograft experiment (collagenated group). In turn, this means that both access windows were prepared using drills.”

and

We have added the following part (lines 412-420):

“Despite efforts to increase the homogeneity of the methodology, various limitations must be mentioned. The dimensions of the two access windows were slightly different: 3.5 × 6 mm (21 mm2) in the non-collagenated group and 4 × 4 mm (16 mm2) in the DBBM group. However, access windows of 18 mm2 and 30 mm2 did not show differences in bone formation in a similar study in rabbits [65]. The different collagen membranes that covered the osteotomies in the two experiments might be considered another limitation of the study. However, it is unlikely that this difference affected the healing of the internal sides of the elevated spaces. This has been substantiated by both animal [66] and human [46] studies.”

“Did the assessor who performed the histological measurements have any calibration training? If so, present the results and the statistical test used.

Authors’ answer.

We have rephrased this paragraph as follows (lines 193-198):

“The surgery in both groups was performed by the same surgeon (E.R.S.; see acknowledgements), and both histological measurements were performed by a well-trained examiner (K.A.A.A.; see acknowledgements). Before starting the measurements, a calibration with another professional (D.B.; see acknowledgements) was per-formed, and it was carried out until the inter-rater agreement in the identification of the tissues achieved a Cohen’s coefficient k > 0.80.”

-. Please indicate the statistical test used to establish the normal distribution of the data, and its result.

Authors’ answer.

We have now applied a Shapiro-Wilk test for the assessment of normality for all variables evaluated and subsequently we have selected the corresponding test, either the unpair t test or the Mann-Whitney test. After the new evaluations, few differences turned to be statistically significant, and we have consequently updated all tables. These are the parameters changed: in the 2-week period, New bone and Xenograft in the Bone wall region; in the 4-week period, only the Soft tissues in the Sub-mucosa region; in the 8-week period no changes were found.

We have rephrased the paragraph as follows (lines 218-220): “The Shapiro-Wilk test for the assessment of normality was applied for all variables. The unpaired t-test or Mann-Whitney test was used for statistical analyses. The significance level was set at p<0.05.”

Results

-. Page 6, lines 197-202. Please present P-values.

-. Page 7, lines 204-207. For clarity present P-values.

-. Page 8, lines 218-220. Please present P-values.

-. Page 8, lines 221-222. Please present P-values.

-. Page 9, lines 233-235. Please present P-values.

Authors’ answer.

We have added the p values in all instances in the text, and we have also added all p values in the 3 tables (lines 226-269).

Discussion.

-. Page 9, lines 241-243. An explanation for that difference should be provided.

Authors’ answer.

We have added the following sentence (lines 355-358):

“Considering the tendency of the sinus to regain its original size [6-8,28,33,34], the higher rate of graft resorption in the collagenated group than in the DBBM group contributed to the greater loss of dimension of the elevated space observed in the former than in the latter.”

-. Page 10, lines 278-280. Present the bases for an increase in blood vessels.

Authors’ answer.

We have implemented this part as follows (lines 359-364):

“In the DBBM group, the proportion of vessels decreased between 2 and 8 weeks, while in the collagenated group, the content in vessels continuously increased, which might in-fluence healing. Blood provides nutrients, gas exchange, immune system cells, mesenchymal stem cells, and growth factors [52-54]. It has been shown that various biomaterials, including xenogeneic grafts, were associated pro-angiogenic effect and an increased expression of various grow factors [53].”

-. The results should be contrasted with more studies on humans.

Authors’ answer.

We have added and discussed some clinical studies as follows (lines 293-299):

“The influence of bone walls on bone formation was also shown in a clinical study [41]. Biopsies were collected from 18 patients six months after sinus floor elevation. A higher percentage of new bone was observed in the mesial sites, closer to the mesial bone wall, than in the distal sites. Moreover, a strong negative correlation between bone percentage and sinus width was found. This, in turn, means that the greater the distance between the lateral and palatal walls, the lower is bone formation.”

And (lines 335-345)

“It has to be considered that the presence of the biomaterial inside the bone (composite bone) might interfere with the osseointegration of implants installed both in a simultaneous [51] or delayed fashion [46-49]. In a rabbit experiment [51], implants were placed simultaneously with sinus augmentation performed with a non-collagenated xenograft. Granules of the biomaterial were found in contact with the implant surface in percentages ranging between 1.7% and 3.6 %. Biopsies of mini-implants installed 6 months after sinus floor elevation were collected from humans after a further 3 months of healing [46-49]. The implant surface was found to be in contact with granules of biomaterials that hampered the growth of newly formed bone on the implant surface. The percentages presented a large range when a collagenated xenograft was used (0.6 to 15.9%) [47-49], but reached 25.1-27.2 when a non-collagenated xenograft was used [46].

And (lines 306-327)

“The percentages of new bone found after eight weeks of healing in the present study were similar to those reported in human histological studies. In a clinical study [45], forty-eight sinuses were elevated with either DBBM or synthetic graft. Biopsies were collected 6–8 months after surgery. The percentages of mineralized bone were 19.8% and 21.6 %, respectively. In another clinical study [46], bovine cancellous bone obtained at a high temperature (> 1200 °C) was used for sinus floor elevation. Mini-implants were installed 6 months after healing and were retrieved after another 3 months. Histological examination of the biopsies revealed 21-23% mineralized bone. In sinuses elevated with a collagenated xenograft similar to that used in the present study, and with mini-implants again in-stalled after 6 months of healing and retrieved after 3 months, percentages of mineralized bone ranging between ~32-39% were found [47-49]. In another clinical study [50], biopsies collected through a crestal approach revealed 40.1 % mineralized bone. The percentages of mineralized bone reported in the above-mentioned studies were similar to those found close to the bone walls in the present study. This outcome might be related to the region from which the biopsies were collected in clinical studies, that is, the bone crest. This region contained the residual bone crest, and it was close to the floor and sinus walls, a condition that resembles that in the experiment in rabbits.

This aspect has important clinical relevance, because the region close to the base of the sinus is where the implants will be installed. Therefore, efforts should be made to pre-serve this region, for example, by making the access windows as small as possible and far away from the sinus floor to increase the source of new bone from the lateral sinus wall [33-34].

-. The study has more limitations that should be mentioned.

Authors’ answer.

We have implemented this paragraph as follows (lines 412-424):

“Despite efforts to increase the homogeneity of the methodology, various limitations must be mentioned. The dimensions of the two access windows were slightly different: 3.5 × 6 mm (21 mm2) in the non-collagenated group and 4 × 4 mm (16 mm2) in the DBBM group. However, access windows of 18 mm2 and 30 mm2 did not show differences in bone formation in a similar study in rabbits [65]. The different collagen membranes that covered the osteotomies in the two experiments might be considered another limitation of the study. However, it is unlikely that this difference affected the healing of the internal sides of the elevated spaces. This has been substantiated by both animal [66] and human [46] studies. The model used should also be considered as a limitation. In fact, when interpreting the results of the present study to humans, the different speeds of healing are an important topic to be considered [67]. Other important limitations are the short period of evaluation and the selection of material from previous studies, even though efforts were made to maintain a high homogeneity between the two studies.”

Conclusions

They should be presented indicating that the study has limitations, and the results should be evaluated with caution.

Authors’ answer.

We have rephrased the conclusion as follows (lines 426-430): “The present study showed that both xenografts allowed new bone formation. In comparison with a non-collagenated xenograft, a collagenated xenograft underwent higher resorption, resulting in greater shrinkage of the elevated space after sinus lifting and a higher content of new bone in the regions close to the bone walls. However, considering the limitations of the present study, the results should be interpreted with caution.”

The language must be edited.

Authors’ answer: We have extensively edited the text.

Reviewer 2 Report

Interesting study, well structured and well executed. Just a few criticisms listed below:

-In the final part of the abstract section insert a sentence on the possible clinical correlations of the study

-check that all keywords are pubmed mesh terms

- in the introduction section, in addition to the amount of bone, the technologies and procedures that can be implemented to improve osseointegration and possibly better sterilize the operating field should also be listed. In this regard, I suggest to insert in the reference section the following scientific work that could be of help:

Valenti C, Pagano S, Bozza S, et al. Use of the Er:YAG Laser in Conservative Dentistry: Evaluation of the Microbial Population in Carious Lesions. Materials (Basel). 2021;14(9):2387. Published 2021 May 4. doi:10.3390/ma14092387

- Some general considerations on the bone healing and regeneration process from a histological point of view should be added in the discussion section, with particular reference to the aspects relating to the lymphocyte populations involved. In this regard, I suggest to insert in the reference section the following scientific work that could be of help to the reader:

Nesti M, Carli E, Giaquinto C, Rampon O, Nastasio S, Giuca MR. Correlation between viral load, plasma levels of CD4 - CD8 T lymphocytes and AIDS-related oral diseases: a multicenter study on 30 HIV+ children in the HAART era. J Biol Regul Homeost Agents. 2012;26(3):527-537.

Author Response

Open Review

English language and style

( ) English very difficult to understand/incomprehensible
(x) Extensive editing of English language and style required
( ) Moderate English changes required
( ) English language and style are fine/minor spell check required
( ) I don't feel qualified to judge about the English language and style

Yes

Can be improved

Must be improved

Not applicable

Does the introduction provide sufficient background and include all relevant references?

( )

(x)

( )

( )

Are all the cited references relevant to the research?

(x)

( )

( )

( )

Is the research design appropriate?

(x)

( )

( )

( )

Are the methods adequately described?

(x)

( )

( )

( )

Are the results clearly presented?

(x)

( )

( )

( )

Are the conclusions supported by the results?

( )

(x)

( )

( )

Comments and Suggestions for Authors

Interesting study, well structured and well executed. Just a few criticisms listed below:

Authors’ answer: we thank you very much the reviewer for the important suggestions. You have added all information requested that have improved the quality of the article. Below you can find all changes made in abstract and text.

Kind regards

The authors

-In the final part of the abstract section insert a sentence on the possible clinical correlations of the study

Authors’ answer: we have added the following sentences at the end of the abstract, while the full explanation of these statements is reported within the text:

“Clinical relevance: In this study, the region adjacent to the bone wall showed the highest new bone content. This region resembles the base of the sinus, closest to the sinus floor and walls, and is the most important region from a clinical point of view, because it is where the implant will be installed. Residues of the biomaterial remained after eight weeks of healing. Other reports have shown that these biomaterial residues may interfere with the integration of implants.”

-check that all keywords are pubmed mesh terms

Authors’ answer: we have selected the following mesh terms:

“Bone Substitutes; Sinus Floor Augmentation; Bone Formation; Bone Regeneration; Bone Resorption; Animal Model”

- in the introduction section, in addition to the amount of bone, the technologies and procedures that can be implemented to improve osseointegration and possibly better sterilize the operating field should also be listed. In this regard, I suggest to insert in the reference section the following scientific work that could be of help:

Valenti C, Pagano S, Bozza S, et al. Use of the Er:YAG Laser in Conservative Dentistry: Evaluation of the Microbial Population in Carious Lesions. Materials (Basel). 2021;14(9):2387. Published 2021 May 4. doi:10.3390/ma14092387

Authors’ answer.

We have added the following part and included the suggested reference (lines 387-392): “The surgical procedures performed in the experiments included in the present study were conducted under sterile conditions, although such conditions could not be completely obtained in the oral environment. Nevertheless, it might help to reduce the possible contamination of instruments, biomaterials, and the operative field [60-61]. Technologies and procedures may be implemented with the aim of better sterilizing the operating field and improving healing [63].

- Some general considerations on the bone healing and regeneration process from a histological point of view should be added in the discussion section, with particular reference to the aspects relating to the lymphocyte populations involved. In this regard, I suggest to insert in the reference section the following scientific work that could be of help to the reader:

Nesti M, Carli E, Giaquinto C, Rampon O, Nastasio S, Giuca MR. Correlation between viral load, plasma levels of CD4 - CD8 T lymphocytes and AIDS-related oral diseases: a multicenter study on 30 HIV+ children in the HAART era. J Biol Regul Homeost Agents. 2012;26(3):527-537.

Authors’ answer.

We have added the following part and included the suggested reference (lines 367-375): “The present study did not include gene expression analysis. Nevertheless, it should be considered that lymphocytes play an important role in the immune system, and deter-mine the specificity of the immune response following infections [55]. Damage to the bone triggers the immune response and the activation of neutrophils, mast cells, monocytes, and macrophages [56]. It has been shown in mice that the absence of lymphocytes B and T compromises bone regeneration in fractures [57]. In an experiment in rabbit femurs [58], gene expression analysis was performed around implants and in sham sites. Upregulation of the immune system and downregulation of bone resorption markers were observed at the implant sites compared with the sham sites.”

Round 2

Reviewer 1 Report

The authors have resolved all the recommendations indicated. The publication of the manuscript in its current version is recommended.